# Chromosome-Scale Genome Assembly and Triterpenoid Saponin Biosynthesis in Korean Bellflower (*Platycodon grandiflorum*)

**DOI:** 10.3390/ijms24076534

**Published:** 2023-03-31

**Authors:** Dong-Jun Lee, Ji-Weon Choi, Ji-Nam Kang, Si-Myung Lee, Gyu-Hwang Park, Chang-Kug Kim

**Affiliations:** 1Genomics Division, National Institute of Agricultural Sciences, Jeonju 54874, Republic of Korea; 2Postharvest Technology Division, National Institute of Horticultural and Herbal Science, Wanju 55365, Republic of Korea

**Keywords:** chromosome-scale genome, *Platycodon grandiflorum*, triterpenoid saponin biosynthesis

## Abstract

*Platycodon grandiflorum* belongs to the Campanulaceae family and is an important medicinal and food plant in East Asia. However, on the whole, the genome evolution of *P. grandiflorum* and the molecular basis of its major biochemical pathways are poorly understood. We reported a chromosome-scale genome assembly of *P. grandiflorum* based on a hybrid method using Oxford Nanopore Technologies, Illumina sequences, and high-throughput chromosome conformation capture (Hi-C) analysis. The assembled genome was finalized as 574 Mb, containing 41,355 protein-coding genes, and the genome completeness was assessed as 97.6% using a Benchmarking Universal Single-Copy Orthologs analysis. The *P. grandiflorum* genome comprises nine pseudo-chromosomes with 56.9% repeat sequences, and the transcriptome analysis revealed an expansion of the 14 *beta-amylin* genes related to triterpenoid saponin biosynthesis. Our findings provide an understanding of *P. grandiflorum* genome evolution and enable genomic-assisted breeding for the mass production of important components such as triterpenoid saponins.

## 1. Introduction

The Korean bellflower (*Platycodon grandiflorum*), a herbaceous, perennial, bell-shaped flower, belongs to the Campanulaceae family. Bellflower is used in traditional medicines and food additives due to its therapeutic effects on bronchitis, sore throats, and asthma [1]. Bellflower root is an oriental herbal medicine used in East Asia, and the flowers are used for ornamental purposes in the USA and Europe [2,3,4]. The applications of *P. grandiflorum* have increased in oriental medicine, food fields, herbal teas, ornamental flowers, and health products [5,6].

In the Campanulaceae family, *Platycodon grandiflorus* A. De Candolle (*P. grandiflorus* A. DC) is the sole species in the Platycodon genus [7]. However, some papers have described *Platycodon grandiflorum* A. De Candolle (*P. grandiflorum* A. DC) as the only species in the Platycodon genus [8,9,10]. In many cases, both scientific names have been used for the same plant, without distinguishing between *P. grandiflorum* and *P. grandiflorus* [11,12]. In the Platycodon genus, the taxonomic delimitation of some species is difficult because historical descriptions are not detailed and the morphological characteristics are similar. The scientific names of *P. grandiflorus* and *P. grandiflorum* are often confused because of an orthographic change in the epithets, which has led to misinterpretations [13]. In this study, *P. grandiflorum* is defined as a variety of *P. grandiflorus* species, because this orthographic error has made it impossible to search databases for information about a single species.

In the *P. grandiflorus* species, two genome sequence assemblies using Chinese balloon flower [14] and Korean balloon flower [15] were reported. Genomic information on these species is available to guide breeding strategies for crop improvement [16]; however, the molecular basis of *P*. *grandiflorum* remains uncharacterized. The recent development of high-throughput sequencing technologies has reduced the difficulty of genomic research, and next-generation sequencing (NGS) technologies such as Oxford Nanopore Technologies (ONT) have enabled accurate genome assembly [17].

Triterpenoid saponins are the most important bioactive components of *P. grandiflorum* [18,19] and have pharmacological activities such as anti-inflammatory, anti-obesity, and hepatotoxic effects [20,21]. Triterpenoid saponins are surface-active glycosides of triterpenes that possess a wide, biologically active group of terpenoids. The triterpenoid saponin of *P. grandiflorum* has a typical oleanane backbone with glucose at the C-30 position and an ester linkage between C-28 and arabinose [22]. The major components are synthesized via mevalonic acid (MVA) and methylerythritol 4-phosphate (MEP) pathways, and the first diversifying step in triterpenoid biosynthesis is the cyclization of 2,3-oxidosqualene, catalyzed by an oxidosqualene cyclase (OSC) [15,23]. In particular, CYP716 and bAS gene families indicate the species-specific selection of key player genes towards the triterpenoid saponin biosynthesis and their functions [15]. Triterpenoid saponins include a large chemical diversity of secondary metabolites with carbon skeletons which undergo various modifications, such as oxidation, hydroxylation, or glycosylation, mediated by cytochrome P450 monooxygenases (CYP450s), glycosyltransferases (UGTs), and other enzymes [24]. Various biochemical pathways have been studied in the Campanulaceae family, including the triterpenoid and flavonoid components [25].

Here, we report a chromosome-scale reference genome sequence for *P. grandiflorum* obtained by combining ONT, Illumina sequence, and high-throughput chromosome conformation capture (Hi-C) technology. Comparative genome analyses illuminated the genome evolution, and the genome-wide analysis revealed the evolutionary adaptation of *P. grandiflorum* which has evolved specifically towards triterpenoid saponins. The present study provides insights into genomic resources that will enable further research into this medicinal plant and expand our understanding of the Campanulaceae family.

## 2. Results and Discussion

### 2.1. Genome Assembly

For the chromosome-scale reference genome of *P. grandiflorum*, 574.7 Mb genome sequences were completed with nine pseudo-chromosomes (Table 1 and Figure 1A). To produce a high-quality genome sequence, we generated approximately 52.7 Gb trimmed Nanopore long-read sequences with an average read length of 4911 base pairs (bp). In addition, 100.5 Gb Illumina short-read length sequences were produced using three insert sizes (Appendix A). The *P. grandiflorum* genome size was estimated to be approximately 634 Mb based on a *k*-mer analysis using Illumina short reads (Appendix A).

To assemble *P. grandiflorum*, firstly, trimmed Nanopore long reads were used to produce the draft assembly after base correction using trimmed Illumina short reads (Appendix A). Secondly, two assemblers were used to compare the accuracy of the draft assembly. Two contig sets were identified by polishing Illumina short-read data (Appendix A). Thirdly, one contig set was selected due to scaffolding efficiency, which was subjected to preliminary scaffolding with Hi-C data (Appendix A). Finally, the contig sequences were scaffolded into nine pseudo-chromosomes using a manual polishing process in the Hi-C analysis (Figure 1B).

The *P. grandiflorum* genome was completed at 574.7 Mb (90.6% of the estimated genome size), being composed of 429 sequence sets (chromosome and contigs) with an N50 value of 64.5 Mb (Table 1). The comprising nine pseudo-chromosomes ranged from 43.5 to 93.4 Mb, and the Hi-C interaction heatmap shows the interaction signals on the nine chromosomes (Figure 1B and Appendix A). The quality of the assembled genome was evaluated using a Benchmarking Universal Single-Copy Orthologs (BUSCO) analysis, which captured 97.6% of the 1614 conserved orthologous angiosperm genes (Table 1). The assembled genome was validated by mapping paired-end (PE) sequences based on three categories. Trimmed PE reads from 95.3% to 96.8% were mapped to the assembled genome sequences, and a maximum of 94.5% of the genome was covered by the mapped reads (Appendix A).

### 2.2. Genome Annotation

A total of 41,355 protein-coding genes were predicted using an evidence-based annotation pipeline (Appendix A) based on transcriptome data (Appendix A) and protein sequences. The genome annotation identified 327.5 Mb repeat sequences in *P. grandiflorum*, accounting for 56.9% of the genome (Table 1). The most repetitive category was long terminal repeat (LTR) retrotransposons, including *Gypsy* (13.0%) and *Copia* (11.8%) (Appendix A). The total length of the protein-coding genes was 40.6 Mb, with an average gene length of 982 bp and a GC content of 44.5%. Among 41,355 protein-coding genes, 80.5% were functionally annotated by comparing their homology against libraries of known proteins, such as NCBI GenBank, protein domains, gene ontology (GO), and the Kyoto Encyclopedia of Genes and Genomes (KEGG). In total, 79.8% of the genes showed high similarity with known protein sequences deposited in NCBI GenBank, and 61.2% of the genes had known conserved protein domains. A total of 41.8% of the genes were assigned to at least one GO term, and 27.6% were assigned to a pathway registered in the KEGG database (Appendix A). For the functional classification of annotated genes, 17,286 GO term genes were classified into functional categories. Within the three categories at level three, the most enriched genes were assigned to organic substance processes, organic cyclic compounds, and membrane processes (Figure 2A).

### 2.3. Genome Comparison

To evaluate genome completeness and annotation accuracy, we compared *P. grandiflorum* and two other *Platycodon grandiflorus* genomes [14,23], all of which belong to the Campanulaceae family. For genome completeness, the BUSCO value was 96.7–98.3% (Appendix A) and the metrics of the annotation gene set were similar to those previously reported for the two *Platycodon grandiflorus* genomes (Appendix A). Therefore, whole-genome sequencing and an annotation comparison indirectly showed that the *P. grandiflorum* genome was well-assembled.

To determine the the evolutionary relationship between *P. grandiflorum* and four other whole genomes (*Panax ginseng*, *Helianthus annuus*, *Solanum lycopersicum*, and *Arabidopsis thaliana*) (Appendix A), we performed a similarity-based gene clustering analysis. A total of 10,282 genes were shared among the five genomes, and 1422 genes were unique to *P. grandiflorum* (Figure 2B). To identify enriched functional categories, we performed GO enrichment analysis using shared and unique genes. In the GO term groups, shared genes were clustered in the regulation of transcription, oxidoreductase activity, and DNA binding. Unique genes were clustered in relation to post-embryonic development (Appendix A), and this GO term is controlled by multiple transcription factors during post-embryonic development [26].

In addition, a GO enrichment analysis was performed using *P. grandiflorum* and two *Platycodon grandiflorus* genomes. Gene clustering revealed that 16,116 genes were shared among the three genomes and 866 genes were unique (Appendix A and Appendix A).

### 2.4. Identification of Genes Involved in Triterpenoid Saponin Biosynthesis

To reveal the expression profile of *P. grandiflorum*, RNA-Seq data identified a gene expressed in at least one of eight different tissues and one of three methyl jasmonate (MJ) treatments. To identify the genes related to triterpenoid saponins in *P. grandiflorum*, 46 candidate genes were identified to be involved in the triterpenoid biosynthesis pathway using the KEGG database (Figure 3). Triterpenoids are generated by OSCs that catalyze the production of diverse triterpene skeletons [24], and *β-amyrin* synthases are important OSCs for triterpenoid saponin biosynthesis [27]. Therefore, the existence of 14 *beta-amyrin* (*β-amyrin*) synthases is evidence of their major contribution to triterpenoid saponin modifications in *P. grandiflorum*. Most *β-amyrin* genes were strongly expressed in roots and seeds (Figure 3). Among the fourteen *β-amyrin* genes, three genes (Pg_chr02_56820T, Pg_chr04_41420T, and Pg_chr04_41430T) showed significantly higher expression in root tissues under normal conditions (Appendix A), suggesting they are key genes associated with the production of triterpenoid saponins in the roots of *P. grandiflorum*. However, two *β-amyrin* genes (Pg_chr04_41450T and Pg_chr04_41400T) significantly expressed in roots were not selected as key genes because of their relatively low expression in roots with the absence of expression in other tissues.

In addition, the expression of Pg_chr02_56820T increased at 12–48 h after exposure after exogenous hormone MJ treatment, indicating its potential association with the induction of triterpenoid saponins [28]. This expression pattern was supported by a previous report [29] related to the efficient induction of ginsenoside biosynthesis in *Panax* species. Considering the high expression of this gene in roots, it may be involved in the accumulation of triterpenoid saponins in roots.

## 3. Materials and Methods

### 3.1. Plant Materials and Genome Sequencing

The whole plant body of *P. grandiflorum* cultivar doraji was collected from the National Institute of Agricultural Sciences research field in Jeonju, Republic of Korea, and was registered at the National Agrobiodiversity Center (http://genebank.rda.go.kr/, accessed on 17 January 2022) under voucher number IT209935. Genomic DNA was extracted from young leaves using the GeneAll Exgene Plant SV kit (GeneAll, Seoul, Republic of Korea) and purified using 0.7x AMPure XP beads (Beckman Coulter, Brea, CA, USA). Nanopore sequencing libraries were constructed using an ONT 1D ligation sequencing kit and sequenced using 1D flow cells (Oxford Nanopore Technologies, Oxford, UK). Nanopore sequencing data (Q > 7) and Illumina PE data (Phred score > 20) were trimmed using Trimmomatic (version 0.38) with the default parameters [30]. During Hi-C sequencing, chromatin conformation capture data were produced by Phase Genomics (Phase Genomics, Seattle, WA, USA) using the Proximo Hi-C Plant Kit (Phase Genomics, Seattle, WA, USA) following the manufacturer’s instructions. The molecules were pulled down with streptavidin beads and used for the construction of an Illumina-compatible sequencing library and sequencing using the HiSeq X platform (Illumina, San Diego, CA, USA).

### 3.2. Genome Size Estimation

The genome size of *P. grandiflorum* was estimated via *k*-mer frequency analyses of high-quality Illumina PE data using Jellyfish (version 2.0) [31] and the GenomeScope tool (version 2.0) [32]. PE data were analyzed using Jellyfish to obtain *k*-mer frequency data using an optimal *k*-mer value of 17. Then, the genome size was estimated using GenomeScope based on the 17 *k*-mer frequency value.

### 3.3. Genome Assembly

First, the trimmed nanopore sequencing data were self-corrected using the Canu assembler (version 1.71) [33] with default parameters to generate the corrected nanopore sequences. The corrected nanopore sequences were de novo assembled using SMARTdenovo [34] and the Canu assembler. The parameters of both assemblers were set with a minimum read length of 1000 bp and the remaining parameters were set to default settings. The assembled contig sequences were polished twice using Pilon (version 1.23) [35] with the trimmed PE data. To improve the quality of genome assembly, additional polishing was performed by mapping the trimmed PE data and generating a consensus sequence, as described in [36]. Haplotigs were removed from the contig sequences using Purge haplotigs [37] with default parameters. In the Hi-C data, scaffolding was performed using Phase Genomics (Phase Genomics, Seattle, WA, USA), and read pairs were aligned using BWA-MEM (version 0.7.17) [38]. The Proximo Hi-C genome scaffolding platform was used to create chromosome-scale scaffolds [39]. The completeness of the gene set predicted in the genome sequence was validated via an analysis using BUSCO (version 5.0.0) [40] with the Embryophyta _odb10 lineage dataset. In addition, genome assembly was further validated by mapping the PE data using BWA-MEM with the default parameters.

### 3.4. Gene Annotation

Protein-coding regions were predicted in genome sequence using an evidence-based annotation pipeline (Appendix A). For transcriptome evidence, RNA-Seq data produced in [23] were downloaded from the NCBI Sequence Rad Archive (SRA) database (https://www.ncbi.nlm.nih.gov/bioproject/526590, accessed on 12 December 2022) and de novo assembled using Trinity [41]. A total of 180,652 transcripts were prepared, with a total length of 184.4 Mb. Protein sequences were compared from three published genome assemblies: sunflower (*Helianthus annuus*, version 1.2), Korean ginseng (*Panax ginseng*, version 1.1), and *Arabidopsis thaliana* (Araport11).

Consensus repeat sequences in the genome sequence were identified and characterized using RepeatModeler (version 1.0.11, http://www.repeatmasker.org/, accessed on 12 September 2022), then combined with known repeat sequences in RepBase (version 28.04, https://www.girinst.org/, accessed on 19 September 2022). The combined sequences were referenced using RepeatMasker (version 4.0.9, http://www.repeatmasker.org/, accessed on 12 September 2022) to identify repeat sequences in the genome. The genome sequences were masked with repeat sequences using an annotation pipeline composed of MAKER3, SNAP, AUGUSTUS, GeneMark-ES, and EvidenceModeler (Appendix A). The final gene set was selected as the gene sequence with an annotation evidence distance (AED) score of <1.

### 3.5. Genome Comparative and Functional Annotation

For a comparative analysis, *P. grandiflorum* and four plant genomes were used: *P. ginseng* (Korean ginseng), *H. annuus* (sunflower), *S. lycopersicum* (tomato), *and Arabidopsis thaliana*. In addition, *P. grandiflorum* and two *Platycodon* species, Korean and Chinese balloon flowers, all belonging to the Campanulaceae family, were compared (Appendix A). Genome sequences of *P. grandiflorum* and other genomes were identified by gene clustering using the OrthoVenn2 web tool [42] with an E-value cutoff of 1 × 10^−5^. Functional annotation of the predicted protein-coding genes was performed using sequence similarities in the NCBI non-redundant protein database using DIAMOND (version 0.9.30.131) [43] with an E-value cutoff of 1 × 10^−5^. Based on these similarity search results, gene ontology (GO) terms were assigned to genes using the Blast2GO Command Line (version 1.4.4) [44] with default parameters. Conserved domains within protein-coding genes were determined using InterProScan (version 5.34) [45] with the default parameters.

### 3.6. Genome Comparative and Functional Annotation

For the gene expression profiling of protein-coding genes, 60 total RNA sequencing data were downloaded from the SRA database (https://www.ncbi.nlm.nih.gov/bioproject/526590), 24 RNA data were derived from eight different tissues (leaves, stems, roots, petals, sepals, pistils, stamens, and seeds) with three replications, and 36 RNA data were derived from methyl jasmonate treatments for control, 12, 24, and 48 h with three replications and three repetitions [23].

The RNA-Seq data were trimmed using Trimmomatic [29] with a Phred score > 20 as a parameter. In addition, contaminated sequences from bacteria and viruses were removed using the BBDuk program (https://jgi.doe.gov/, accessed on 29 September 2022) with the *k*-mer 31 parameter. Expression profiles were determined from trimmed RNA reads using HISAT2 (version 2.2.0) [46] with spliced alignment options. The RNA reads were mapped to protein-coding sequences using HTSeq-count (version 0.12.4) [47]. Fragments per kilobase of transcript per million mapped reads (FPKM) were calculated based on the number of mapped RNA reads, and a heatmap was generated using the R package pheatmap [48]. To investigate triterpenoid saponin biosynthesis, candidate genes were identified by searching for genes assigned to the triterpenoid biosynthesis pathway. A metabolic pathway was assigned to genes using the KAAS server [49] on eukaryote gene sets using the single-directional best hit method.

## 4. Conclusions

*P. grandiflorum* is a popular medicinal and health food resource with potential therapeutic effects used throughout East Asia. In this study, we report the chromosome-scale genome assembly and genomic characterization of *P. grandiflorum*. Additionally, transcriptomic analyses were performed to explore the formation of triterpenoid saponins. Our findings provide insights into *P. grandiflorum* genome evolution and enable the future genomic-aided breeding of important traits in the Campanulaceae family. These genomic resources represent important first steps towards the genetic improvement of *P. grandiflorum* for the utilization of plant pharmaceutical resources and will accelerate improvements in agricultural traits, such as the production of triterpenoid saponins, which is the most important bioactive component of *P. grandiflorum*.

## Figures and Tables

**Figure 1 ijms-24-06534-f001:**
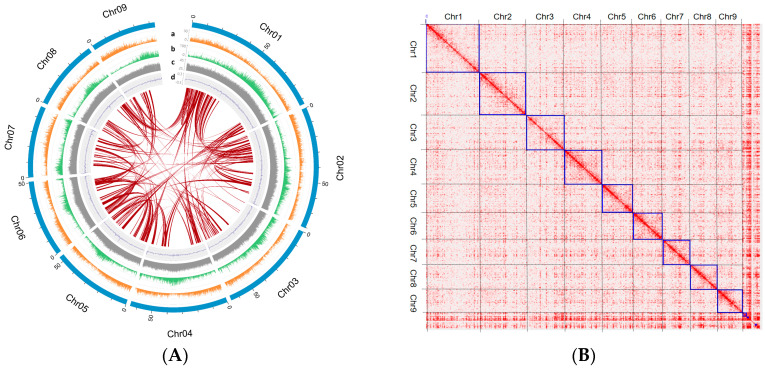
Overview of chromosome scale in *P. grandiflorum*. (**A**) Circos plot of the nine pseudo-chromosomes. The outermost track (light blue) indicates nine pseudo-chromosomes. The a–d track shows the distribution of genes (a, orange), repeat distribution (b, green), GC contents (c, gray), and GC skew (d, blue) in non-overlapping 100 kb sliding windows. The innermost lines (red) indicate collinear blocks identified within the pseudo-chromosomes. (**B**) Heatmap of Hi-C assisted assembly. Blue box indicates higher contact probability.

**Figure 2 ijms-24-06534-f002:**
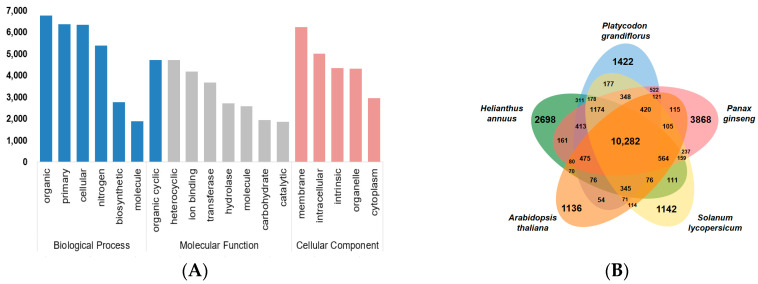
Gene distribution of *P. grandiflorum*. (**A**) Gene ontology (GO) classification of genes. The 17,286 genes assigned to GO terms were classified in functional categories based on GO terms at level 3. The *Y*-axis is the number of genes. (**B**) Shared and unique gene clusters in *P. grandiflorum* and four related species. Venn diagram illustrating the number of shared and unique gene clusters among *P. grandiflorum* (Korean bellflower), *P. ginseng* (Korean ginseng), *H. annuus* (sunflower), *S. lycopersicum* (tomato), and *Arabidopsis thaliana*.

**Figure 3 ijms-24-06534-f003:**
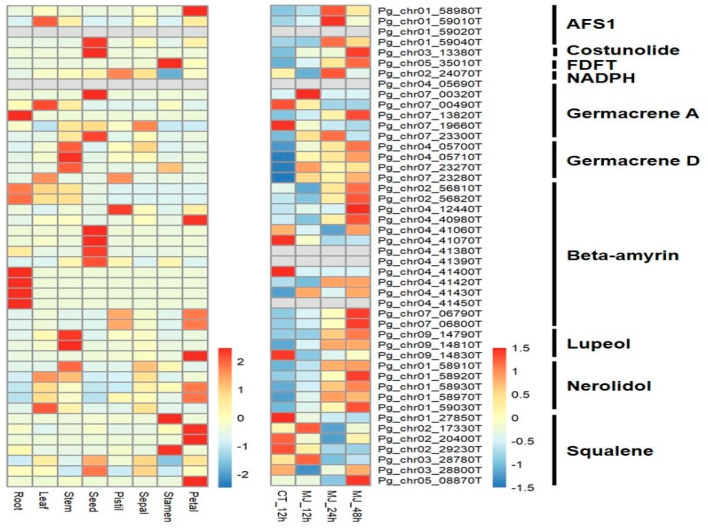
Expression patterns of candidate genes involved in the triterpenoid biosynthesis pathway in eight different tissues and three methyl jasmonate (MJ) treatments. Expression values were scaled per row (i.e., per gene) to visualize gene expression peaks among the different samples. Sample names and gene symbols are shown at the bottom and on the right of the heatmap, respectively. RNA sequencing data taken from the SRA database (https://www.ncbi.nlm.nih.gov/bioproject/526590, accessed on 12 December 2022). AFS1: alpha-farnesene synthase, FDFT: farnesyl-diphosphate farnesyltransferase, NADPH: NAD^+^-dependent farnesol dehydrogenase.

**Table 1 ijms-24-06534-t001:** Genome assembly and gene annotation of *P. grandiflorum*.

Parameters	Value
Genome assembly	
Number of sequences	429 *
Total length of sequences	574,706,410 bp
N50 length	64,545,416 bp
Average length	1,339,642 bp
Smallest sequence	1071 bp
Longest sequences	93,483,513 bp
Complete BUSCO	97.6%
Gene annotation	
Number of protein-coding genes	41,355
Total length of protein-coding genes	40,602,816 bp
Average gene length	982 bp
Smallest gene length	102 bp
Longest gene length	15,303 bp
Repeat content	56.9%
GC content	44.5%
Functionally annotated	80.5%
Complete BUSCO	86.5%

* Nine pseudo-chromosomes and 420 contigs.

## Data Availability

All raw sequencing data produced in this study were deposited into the NCBI Sequence Rad Archive (SRA) under BioProject number PRJNA656905 and BioSample SAMN15806942. This Whole Genome Shotgun project has been deposited in GenBank under accession number JACONP000000000 under BioProject number PRJNA656905. In addition, files for the annotated gene sequences and functional annotations were uploaded to Figshare (https://doi.org/10.6084/m9.figshare.21511020).

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
