# Peer review of "Chromosome-Scale Genome Assembly and Triterpenoid Saponin Biosynthesis in Korean Bellflower (Platycodon grandiflorum)"

_ijms, 2023, doi:10.3390/ijms24076534_

Round 1

Reviewer 1 Report

The manuscript has clear goals (genome assembly\annotation\prediction of candidate genes), which were achieved using appropriative methods. The results are clearly presented, and I think the manuscript can be accepted in its current form.

However, I have some comments which may improve the final version:

1) Heat map in Figure 3. According to it, some β-amyrin genes (for instance, Pg_chr04_41450T, Pg_chr04_41400T) seem to be significantly expressed in roots. But not discussed in the text. Only when you open the Supplementary file, it becomes clear why it is so (relatively low expression in roots and the absence of expression in other tissues). Thus it may be distractive for readers. Therefore, I recommend making some changes. For instance, the expression level may be presented as read counts (raw or normalized in median-of-ratio method as described in Anders and Huber (DESeq/DESeq2)). As an alternative p-value or q-value (FDR Benjamini and Hochberg or FDR Storey and Tibshirani) should be added.

2) Table 1. "Genome assembly", "Gene annotation" may be centered in both columns

I hope you will continue future genomic research of P. grandiflorum to improve the current assembly version. The median length of ONT reads is a bit low. And due to the massive amount of repeat regions (a common feature of flowering plants some mistakes (rearrangements) may be detected in future investigations. For instance, making a genetic map of P. grandiflorum or additional PacBio CCS (ONT) sequencing with 20-30 kb median length may allow the improvement of the genome assembly. 

Author Response

Dear reviewer,

Thanks for your attention and received comments of our manuscript. We agree with all comments and all have been corrected in the manuscript files.

Sincerely Yours,

Chang-Kug Kim

Reviewer 2 Report

This manuscript present detailed analysis of the genetic basis for particular type of triterpenoid saponins of a taxon named as Platycodon grandiflorum. The methods used for genome assembly and annotation were those used in the published paper in Journal Horticulture Research (https://doi.org/10.1038/s41438-020-0329-x) for the species Platycodon grandiflorus. In addition, in this published paper were published the original data of the transcriptome, that were referred in this submitted manuscript in order to identify the pathway of particular triterpenoid saponins. Later, other published article also carried out with the species Platycodon grandiflorus (Front. Genet., 08 April 2022 Sec. Plant Genomics; Volume 13 - 2022 https://doi.org/10.3389/fgene.2022.869784); published the first Chromosome-Level Reference for this species, also in this article its genome was assembled and annotated following similar workflow to that published in 2020. In particular, The submitted manuscript to ijms followed similar workflow to that used in this previous published article.

During the review was confusing if the species Platycodon grandiflorus is in fact the studied taxon, and not Platycodon grandiflorum. I think this point can be easily resolved in authors include the taxonomic authority. Since, the two past published articles are very similar to this submitted manuscript, not only in the claims but also in the analytical and methodological approximations; I suggest the following tips in order to strongly mark the differences:

1) In the Introduction say clearly the taxonomic names of that species of Platycodon whose assembled and annotated genome(s) was previously published. Include data if your studied species is in fact a variety of P. grandiflorus, and to avoid confusions add the taxonomic authority in the name of your studied species Platycodon grandiflorum.

2) In the introduction emphasize the findings of those two articles to avoid confusions with your work. For example, the article published in 2020 focused to investigate the genetic bases of the families CYP716 and bAS, however, in it was mentioned that also the genes involved in others triterpenoid saponins (Figure 2 of this published article) were investigated. For example, TSB pathways, including MVA, MEP, isopentenyl pyrophosphate, and OSC pathways (Supplementary Table S1). I suggest that authors clearly establish if your studied molecules already have preliminary results in past published studies. For example, the Figure 2 of the article 2020, seems that in fact contain part of your results.

3) In your objectives explicitly say the names of your studied species and clearly say if you want to identify genetic basis and or the pathways of these molecules.

4) For me, was really confused that in the section of Results and Discussion many methods are named. I marked with red line, colored lines and comments. All those procedures that allowed to obtain the results are in fact methods, moreover, since authors followed methods previously established these could be cited.

5) I found also confusion in origin of the data here analyzed. I suggest, that clearly authors establish those original data produced in this study and cite properly those data that were taken from NCBI site. Take care with this point, do no omit to cite clearly in the text and in the supplementary materials the origin of the data analyzed. For example, you can annotate the legend “de novo sequenced”, or data taken from XXXX.

Author Response

(The authors gave the same response as above.)

Reviewer 3 Report

Comments and Suggestions for Authors

Recommendations: Minor  Revisions

“Chromosome-level genome assembly and triterpenoid saponins biosynthesis in Korean bellflower (Platycodon grandiflorum)

Abstract

Please revised the abstract for clarity. There are some sections in the abstract that are need to modified, and that I believe the authors could communicate better especially the methods sections need more details .

Introduction

Need to modify: There is no mention of the taxonomic status of this genus in the regions. Is this very important. Please identify the relevant literature for more evidence

Line 30-312: The triterpenoid saponin? Please before this need to tell us the importance of this species after ward explicit regarding which chemical constitutes are talking about…

-Introduction still need more modifications such as the introduction part  for taxonomic perspective is very week and they not tell us about

Last two Paragraph: Reword this first few sentences of the paragraph, it is a bit vague.I suggest you provide a cleared background of the study area and its importance.

Methodology

Overall, in the opinion of, this section does not adequately describe the linked to the previous studies. This manuscript needs to do at least two things in order to link with previous studies.

Results and discussion

I have a few general comments on this section  during this round:

The paper is very significant finding, however the figures need to modified and for the analysis and current finding  I will go through it again after second round.

The text provided throughout the results and discussion section is too limited. The authors must expand the discussions of each of the part and reference specific panels of their figures.

 After solving the above problems, I review  this manuscript for further modifications .

Author Response

(The authors gave the same response as above.)

Reviewer 4 Report

To,

The Editor,

IJMS, MDPI,

Manuscript ID: ijms-2288054

Subject: Submission of comments on the manuscript in “IJMS"

Dear Editor IJMS, MDPI,

Thank you very much for the invitation to consider a potential reviewer for the manuscript (ID: ijms-2288054). My comments responses are furnished below as per each reviewer’s comments. 

 Dear Chief Editor,

The reviewed manuscript authors carried out the genome sequence of Platycodon grandiflorum. The assembled genome was finalized as 574 Mb, containing 41,355 protein-coding genes. Genome completeness was assessed as 97.6% by BUSCO analysis. The P. grandiflorum genome comprises nine pseudo-chromosomes with 56.9% repeat sequences. Gene evolution has revealed a significant expansion of the beta-amylin family related to triterpenoid saponin biosynthesis. Our findings provide an understanding of species-specific evolution towards triterpenoid saponin biosynthesis in P. grandiflorum, thus helping to improve its medicinal value through genomic information. Therefore, it might be conditionally accepted as subject to major revision. Instead, authors have to improve their manuscripts with many non-clear meanings, inaccuracies, and the authors need to address the following issues before it can be accepted for publication.

  1. I have read the entire manuscript and my initial comment is that manuscript is poorly written. I have significant concerns about the grammar and vocabulary of the manuscript; therefore, I recommend the authors to used an English proofreading service.
  2. The structure of the abstract should be improved, as well as the lack of several aspects that should be included in this section. Most of the abstracts contain confusing and uninformative sentences. Please give more precise objectives here (such as in the Abstract). The abstract should highlight the most important results of the parameters and characteristics assayed.
  3. Introduction grammatical issues appear to be most prevalent in the introduction, making for very confusing reading. Further, the introduction is short but has no clear thread.
  4. Why you selected this crop for your experiment? Please provide the detail of the used variety.
  5. The figures are quite low resolution and difficult to make out. Higher-resolution versions will be needed for publication. Further, text in figure is not readble. for example, in Figures 1, 2, 3, 4, and 5.
  6. The discussion should be interpreted with the results as well as discussed in relation to the present literature.
  7. Author must add the conclusion section.
  8. In Material and Methods:- indicate how many replicates assayed in each analysis/parameter. The number of samples or biological and technical replicates should be mentioned for each parameter in the methods.
  9. References: shall have to correct the whole References according to the ”Instructions for the Authors”, e.g. the Journal name must be abbreviated, journal name in italics, the year must be bold and you shall have to use the abbreviated number of the Journals cited. Further, some references the title to paper in title case some are in small letter case, hence please follow the journal instruction. Moreover, the scientific name must be italics. Please check the all refernce carefully.
  10. Line no. 306 Lactuca sativa must be italic.

Best wishes

Author Response

(The authors gave the same response as above.)

Round 2

Reviewer 4 Report

Dear Editor,

Thank you for providing the opportunity to review the revised manuscript. The manuscript is improved considerably after revision according to the reviewer's comment. Now this study is a suitable contribution to the IJMS. I recommend the manuscript for publication.

Thank you

With best regards